# Augmenting Azoles with Drug Synergy to Expand the Antifungal Toolbox

**DOI:** 10.3390/ph15040482

**Published:** 2022-04-14

**Authors:** Aidan Kane, Dee A. Carter

**Affiliations:** School of Life and Environmental Sciences and Sydney ID, University of Sydney, Camperdown, NSW 2006, Australia; aidan.kane@sydney.edu.au

**Keywords:** antifungal, azole, synergy, mycosis, resistance, *Candida*, dermatophytes, natural products

## Abstract

Fungal infections impact the lives of at least 12 million people every year, killing over 1.5 million. Wide-spread use of fungicides and prophylactic antifungal therapy have driven resistance in many serious fungal pathogens, and there is an urgent need to expand the current antifungal arsenal. Recent research has focused on improving azoles, our most successful class of antifungals, by looking for synergistic interactions with secondary compounds. Synergists can co-operate with azoles by targeting steps in related pathways, or they may act on mechanisms related to resistance such as active efflux or on totally disparate pathways or processes. A variety of sources of potential synergists have been explored, including pre-existing antimicrobials, pharmaceuticals approved for other uses, bioactive natural compounds and phytochemicals, and novel synthetic compounds. Synergy can successfully widen the antifungal spectrum, decrease inhibitory dosages, reduce toxicity, and prevent the development of resistance. This review highlights the diversity of mechanisms that have been exploited for the purposes of azole synergy and demonstrates that synergy remains a promising approach for meeting the urgent need for novel antifungal strategies.

## 1. Introduction

### 1.1. The Burden of Fungal Disease

Fungal pathogens present an ever-increasing threat to global health. An estimated 1.5 million people are killed by fungal infections every year, and the incidence of several serious mycoses is growing [1,2]. It is likely that the global fungal burden is under-estimated, as several invasive fungal infections are under-reported in the developed world due to their association with other predisposing illnesses [3,4,5]. Australian clinics have recently seen a near-doubling of systemic candidaemia caused by drug-resistant *Candida glabrata*, with increasing rates of invasive candidaemia seen in Europe and the USA [6,7,8]. *Candida auris*, an emerging yeast pathogen infamous for its high tolerance to most important antifungals, has been the cause of several recent outbreaks, both before and during the COVID-19 pandemic [9,10,11,12]. Drug-resistant biofilms of various species of *Candida* have become increasingly responsible for fatal nosocomial infections [13]. Lethal infections with *Aspergillus fumigatus* and *Cryptococcus* sp. also remain a pressing concern, together causing an estimated 400,000 deaths per year, with chronic pulmonary aspergillosis severely affecting close to 3 million people [2]. Morbidity of non-life-threatening topical fungal infections is also increasing at an alarming rate. Cutaneous dermatophytosis affects 12 to 13 million people a year, and nail infections are extraordinarily difficult to treat, with less than 13% of cases fully resolved [14,15]. Decreasing susceptibility to topical antifungals has been observed in *Exophiala dermatitidis* and *Malassezia* sp. [15,16,17,18].

The increased incidence of emerging systemic, superficial, and cutaneous fungal pathogens has increased the demand for novel antifungal medications. However, despite the advances over the past four decades in bringing azoles and echinocandins to market, the currently available antifungals still operate via a limited number of mechanisms (Figure 1) and unfortunately all have significant problems, including toxicity, difficulty of administration, limited bioavailability and efficacy, and an often high cost [19,20,21].

Current therapies are also being increasingly compromised by antifungal resistance. Widespread over-use of agricultural fungicides appears to be driving cross-resistance in a range of pathogenic moulds and yeasts, and the use of prophylactic and over-the-counter azoles for superficial infections can promote the acquisition of resistance [22,23,24,25]. While recent innovations in anti-retroviral therapy have dramatically reduced the rates of fungal infection associated with HIV-AIDS, the case fatality rates of invasive fungal infections have remained constant [2,26,27]. Due to its propensity to up-regulate active efflux pumps, *C. glabrata* is an emerging cause of recurrent candidiasis, and multiple-drug resistance has been observed in an alarming proportion of clinical isolates of *C. auris* [28,29,30]. Clearly, there is an urgent and unmet need for new approaches to treat fungal infections [31].

### 1.2. Azole Antifungals

Arguably the most successful class of drug in the antifungal toolbox is the azoles. Azoles operate by binding to and inhibiting lanosterol 14α-demethylase, a fungal cytochrome P450 enzyme essential to the biosynthesis of ergosterol (Figure 1) [32]. First-generation triazoles, such as itraconazole and fluconazole, have low host toxicity, and the more important azoles are administered orally and have excellent pharmacodynamic properties [33]. Itraconazole and the second-generation triazole voriconazole have high respiratory bioavailability and fluconazole has superb neural pharmacokinetics, making it well suited as a maintenance therapy for cryptococcal meningitis. A promising new azole prodrug, isavuconazonium, was approved by the FDA in 2015, while a new generation of tetrazoles is currently in clinical and agrochemical development [34,35,36,37,38,39]. Older azoles like fluconazole are cheap, off-patent and widely available, including in developing countries where they are needed the most. The currently available prescription and over-the-counter azoles, and the mycoses for which they are commonly indicated, are detailed in Table 1.

Azoles are a vital tool in the fight against fungal infection. It has been projected that the antifungal market will be USD 13 billion by 2026, with roughly 2.4 billion of that allocated to azoles: USD 1.4 billion in oral and intravenous drugs, and USD 1 billion in topical and over-the-counter solutions [40,41]; however, azoles have limitations. Many filamentous fungal pathogens are not sensitive to azoles, particularly fluconazole [33]. Second-generation triazoles like voriconazole can be toxic to the host [42]. Azoles are often used prophylactically in neutropenic and transplant patients, but this can encourage the development of resistance [23,43]. Azole-based fungicides are used in agriculture on a massive scale, and cross-resistance between these and azole antifungals is seen with alarming frequency [22]. *Candida auris* particularly represents an escalating threat to the usefulness of azoles, as it is frequently highly azole-tolerant and often multi-drug resistant [44].

**Figure 1 pharmaceuticals-15-00482-f001:**
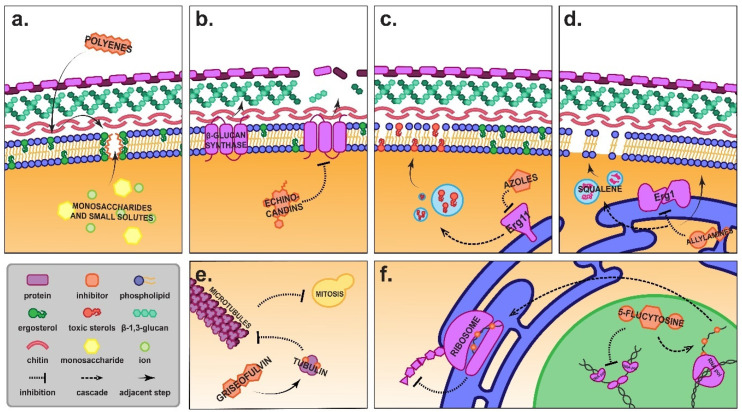
**Mechanisms of action of commercially available antifungals**. (**a**) Polyenes bind to ergosterol in the plasma membrane, forming pores that permit the efflux of vital small solutes like potassium ions and simple sugars [45]. (**b**) Echinocandins operate by inhibiting 1,3-β-D-glucan synthase in the fungal membrane, depriving the cell wall of glucans and therefore its structural integrity [46]. (**c**) Azoles inhibit Erg11 (lanosterol 14α-demethylase) preventing the biosynthesis of ergosterol and resulting in a build-up of toxic methyl-sterols that incorporate into the plasma membrane. The result is a loss of membrane structure and inhibition of growth [47]. (**d**) Allylamines inhibit ergosterol biosynthesis by antagonising squalene epoxidase Erg1, which converts squalene to squalene epoxide. As well as preventing the biosynthesis of ergosterol, this results in the build-up of squalene, which is deposited into lipid vesicles that disrupt the plasma membrane [48]. (**e**) Griseofulvin binds to tubulin in the fungal cell, preventing the formation of microtubules and arresting mitosis [49]. (**f**) 5-flucytosine is a pyrimidine analogue that is converted to 5-fluorouracil inside the cell. This fluoridated nucleotide is incorporated into mRNA, halting ribosomal processing and inhibiting protein translation. 5-fluorouracil also antagonises Cdc21, or thymidylate synthase, preventing the biosynthesis of thymidine nucleotides and inhibiting DNA synthesis [50].

There are various mechanisms of resistance to azole antifungals. Mutations in *ERG11* (also known as *CYP51*), which encodes the target enzyme lanosterol 14α-demethylase, can prevent enzyme binding, or *ERG11* expression can be up-regulated via changes to its promoter or regulating transcription factors or via gene duplication and aneuploidy [51]. Azoles can be actively excluded from the cell via membrane bound ABC and MFS efflux pumps [52,53,54], which can be increased in expression via aneuploidy or by alterations to transcription factors leading to constitutive expression. Mutations in ERG3 have also been found to increase resistance, thought to be via prevention of the build-up of toxic intermediates [55,56,57,58,59,60,61]. Frequently, these resistance mechanisms lead to cross-resistance across azole drug types, and although the research community is working to derive new azole antifungals by modelling lanosterol 14α-demethylase crystal structures [62,63,64], the increased resistance seen in clinical strains can undermine their use as a monotherapy.

**Table 1 pharmaceuticals-15-00482-t001:** Currently available azole antifungals and associated mycoses.

Class	Application	Azole	Brand	Mycosis	Notes	Ref.
**Imidazole**	**Topical**	butoconazole	Gynazole-1, Mycelex-3	uncomplicated and recurrent vaginal candidiasis		[65]
		climbazole	Squaphane, Pitiren	dandruff and seborrhoeic dermatitis caused by *Malassezia* sp.		[66]
		clotrimazole ^†^	Lotrimin	oral and vaginal candidiasis, and tinea versicolor, cruris and pedis	WHO Essential Medicine	[66,67]
		eberconazole	Ebernet	cutaneous candidiasis and dermatophytosis	Approved in EU in 2015	[68]
		econazole	Spectrazole, Ecostatin	tinea pedis and cruris, vaginal candidiasis	Also repels clothes moths	[69]
		flutrimazole	Flusporan, Topiderm	cutaneous dermatophytosis including tinea pedis		[70]
		isoconazole	Icaden, Travogen	tinea pedis and vaginal candidiasis	Effective against Gram-positive bacteria	[71]
		ketoconazole ^†^	Nizoral	seborrhoeic dermatitis, dandruff, tinea and cutaneous candidiasis	Also systemic	[72]
		luliconazole	Luzu	tinea pedis and cruris and other dermatophytoses	FDA-approved in 2013	[73]
		miconazole ^†^	Monistat, Desenex	dermatophytosis and cutaneous, oral and vaginal candidiasis	WHO Essential Medicine	[74]
		oxiconazole	Oxistat, Oxizole	dermatophytoses and cutaneous candidiasis		[75]
		sertaconazole	Ertaczo, Dermofix	tinea pedis and vaginal candidiasis	Also anti-inflammatory and anti-pruritic	[76,77]
		sulconazole	Exelderm	dermatophytoses	Also anti-carpet beetle	[78,79]
		tioconazole	Vagistat-1	onychomycosis, dermatophytoses and vaginal candidiasis	Also called thioconazole	[80]
	**Systemic**	ketoconazole	Nizoral (oral)	mycoses caused by *Candida, Histoplasma* and *Coccidioides*	Systemic use for extreme cases only	[81]
**Triazole**	**Topical**	efinaconazole	Jublia, Clenafin	onychomycosis	Low cure rate, but higher than other drugs	[82]
		fluconazole ^†^	Diflucan	dermatophytoses and cutaneous candidiasis	WHO Essential Medicine, more commonly systemic	[21]
		terconazole	Terazol	acute and chronic vaginal candidiasis		[83]
	**Systemic**	fluconazole ^†^	Diflucan	candidiasis, cryptococcosis, histoplasmosis, blastomycosis	WHO Essential Medicine, oral or intravenous	[21]
		fosfluconazole	Prodif	prophylaxis in the immunocompromised	Fluconazole prodrug	[84,85]
		fosravuconazole	Nailin	onychomycosis	Ravuconazole prodrug	[86]
		isavuconazonium	Cresemba	mucormycosis and invasive aspergillosis	Isavuconazole prodrug	[35,87]
		itraconazole ^†^	Sporanox, Orungal	aspergillosis, histoplasmosis, coccidioidomycosis and blastomycosis	WHO Essential Medicine	[48,88]
		posaconazole	Noxafil, Posanol	invasive candidiasis, aspergilosis, mucormycosis and scedosporiosis	FDA-approved in 2006	[89]
		voriconazole ^†^	Vfend	aspergillosis, candidiasis, penicilliosis, histoplasmosis and fusariosis	WHO Essential Medicine	[90]

^†^ Most commonly prescribed azole antifungals. Other azole antifungals no longer on the market include ravuconazole, a triazole similar to voriconazole which was discontinued after Phase-III clinical trials and the thiazole abafungin, which is no longer available [91].

### 1.3. Antimicrobial Synergy

Recent expiries in patent protection of early-generation azoles have led to generic alternatives for voriconazole, fluconazole, posaconazole and efinaconazole entering the market and have made space for innovations. There is a particular increasing interest in enhancing the antifungal activity of current azoles using drug synergy.

Synergy occurs when two compounds produce an increased inhibitory effect beyond what would be expected by adding the effects of the compounds individually [92]. Significant synergy is determined by the Fractional Inhibitory Concentration Index (*FICI*), which is calculated as the sum of the ratios of the Minimum Inhibitory Concentration (*MIC*) of the drugs when used alone and together, according to the following equation:$$FICI=\frac{MIC\;of\;drug\;A\;in\;combination}{MIC\;of\;drug\;A\;alone}+\frac{MIC\;of\;drug\;B\;in\;combination}{MIC\;of\;drug\;B\;alone}$$

When the *FICI* of two drugs is ≤0.5 their interaction is considered synergistic, and when this is >4 it is considered antagonistic. An *FICI* between 0.5 and 4 is considered indifferent [92].

Antimicrobial synergy can overcome resistance and lower inhibitory dosages to within clinically achievable levels [93,94]. Synergy can expand the spectra of activity of the individual drugs, making azoles a viable option against pathogens that may be otherwise resistant. Antimicrobial synergy can also make otherwise fungistatic drugs fungicidal, including many azoles [95,96].

Antimicrobial synergy has been exploited in the clinic for years for the treatment of HIV and malaria [97,98], and the most successful induction therapy for cryptococcal meningitis is a synergistic combination of amphotericin B and 5-flucytosine. Synergy between two drugs can be potentiated by their co-operation on multiple enzymes belonging to the same pathway; for example, sulfamethoxazole and trimethoprim are antibacterial agents that target the folic acid biosynthesis pathway at two different sites to synergistically improve both the spectrum and potency of bacterial inhibition [99]. Synergy can also be produced between drugs that inhibit different processes: for example, amphotericin B damages the fungal plasma membrane and 5-flucytosine interrupts translation and replication [100,101].

### 1.4. Aims and Scope of This Review

The mechanism of action of azoles is well understood (Figure 1) and various synergists have been described that operate on related (Figure 2) or quite disparate (Figure 3) pathways or mechanisms. In this review, we describe recent studies that have reported azole synergists. There are several excellent earlier reviews that have considered aspects of azole synergy [52,102,103]; here we provide an update with a particular focus on developments made over the past six years. Table 2 provides an overview of these findings presented as a heatmap, with the extent of synergy demonstrated between commonly used azoles and secondary agents in blue, and the spectrum of fungi found to be affected in yellow. Below we explore each of the different classes of synergists and their potential as combination therapies. 

**Table 2 pharmaceuticals-15-00482-t002:** Interactions between azoles and synergists and their spectrum of activity described in published studies.

		Azole Synergy							Synergy in % Strains Tested							
Category	Synergist	Fluconazole	Itraconazole	Voriconazole	Isavuconazole	Posaconazole	Ketoconazole	Miconazole	*C. albicans*	AR *Candida*	*A. fumigatus*	AR *Aspergillus*	Dermatophytes	Other	Notes ^1^	Ref.
**Antimicrobials**																
**Antifungals**	**terbinafine**														Incl. *Scedosporium* sp. and *Pythium* sp.	[104,105,106,107,108,109,110,111,112]
	**caspofungin**														Dep. on mechanism of resistance	[113,114,115,116]
	**anidulafungin**															[114,117,118]
	**micafungin**														Incl. *C. auris*	[115,116,118]
	**natamycin**														Incl. *Fusarium* sp.	[119]
	**ciclopirox**															[120]
	**flucytosine**														Incl. *C. auris*	[121]
	**voriconazole**														Limited synergy in the Mucorales	[122]
	**amorolfine**														Onychomycosis clinical trials	[123,124,125,126]
	**K20**														Also clotrimazole, incl. *C. neoformans*	[127]
	**oxadiazole pept.**															[128]
**Antibacterials**	**sulfamethoxazole**														Incl. *C. auris*	[129,130]
	**sulfa- antibiotics**														Some effective vs. biofilms and in vivo	[130]
	**doxycycline**														Effective against biofilms and in vivo	[131,132,133]
	**tigecycolline**														Incl. *Fusarium* sp., limited anti-biofilm	[132,134,135]
	**minocycline**														Incl. *C. neoformans, Scedosporium* sp.	[133,136,137,138,139]
	**gentamicin**														Effective against biofilms	[140]
	**linezolid**														Little synergy, but reduced dosage	[141]
	**polymyxin B**														Incl. *Rhodotorula* and *Lichtheimia* sp.	[142]
	**colistin**														Incl. *C. auris*	[96,143,144]
**Antiparasitics**	**pyrvinium pam.**															[145,146]
	**chloroquine**															[147]
	**artemisinins**														Effective against biofilms	[146]
	**INK128**														Incl. *Fusarium* and *Exophiala* sp.	[148]
	**mefloquine**														Incl. *C. neoformans*	[149]
**Antivirals**	**saquinavir**														Incl. *Histoplasma capsulatum*	[150]
	**ritonavir**														Incl. *Histoplasma capsulatum*	[150]
	**adamantanamine**														Switch from fungistatic to fungicidal	[151]
	**ribavirin**														Effective against biofilms and in vivo	[152]
	**lopinavir**														Incl. *C. auris*	[153]
**Efflux Inhibitors**																
**Calcium Inhibitors**	**tetrandrine**														Effective in vivo	[154,155,156,157]
	**verapamil**														Dep. on mechanism of resistance	[158,159]
**Other**	**eucalyptal D**														Natural product	[160]
	**dodenoic acid**															[161]
	**azoffluxin**														Incl. *C. auris*	[162]
	**ospemifeme**														Incl. *C. neoformans* and *C. auris*	[163]
	**phialocephalarin**															[164]
	**palmarumycin P3**															[164]
	**geraniol**														Effective in vivo, natural product	[165]
**Repurposed Drugs**																
**Statins**	**lovastatin**														Incl. *Rhizopus* sp.	[166,167,168]
	**atorvastatin**														Incl. *Rhizopus* sp., *Cryptococcus* sp.	[166,167,169]
	**fluvastatin**														Incl. *Rhizopus* sp.	[166,167]
	**simvastatin**														Incl. *Rhizopus* sp., *Cryptococcus* sp.	[166,167,170]
	**pitavastatin**															[171]
**Bisphosphonates**	**risedronate**														Incl. *Cryptococcus* sp.	[172]
	**alendronate**														Incl. *Cryptococcus* sp.	[172]
	**zoledronate**														Incl. *Cryptococcus* sp.	[172]
**Immunomodulators**	**promethazine**															[173,174]
	**terfenadine**														Effective against biofilms	[175]
	**ebastine**														Effective against biofilms	[175]
	**dexamethasone**														Effective against biofilms	[176]
	**budesonide**														Effective in vivo	[177]
	**methotrexate**															[178]
**Psychoactives**	**bromperidol**															[179]
	**fluoxetine**														Effective in vivo, *C. albicans* only	[180]
	**haloperidol**															[173]
	**sertraline**														Incl. *Trichosporon asahii*	[181]
**Calcineurin Inhibitors**	**cyclosporine**														Incl. *S. cerevisiae*	[182,183,184,185,186,187]
	**tacrolimus**														Incl. *S. cerevisiae*	[183,188,189,190,191,192,193]
**Other**	**PPIs**														Proton pump inhibitors	[194]
	**geldanamycin**															[195]
	**ponatinib**														Incl. *C. neoformans*	[196]
	**HSP990**														Incl. *C. neoformans*	[197]
	**givinostat**															[198]
	**lonafarnib**														Incl. *E. dermatitidis*	[199]
	**isoquercitrin**															[200]
	**EDTA**														Incl. *C. deuterogattii*	[201]
	**D-penicillamine**														Copper ion chelator	[202]
	**licofelone**														Effective against biofilms	[203]
	**phenylbutyrate**															[204]
	**17-AAG**														Incl. *E. dermatitidis*	[205]
	**ketamine**															[206]
	**ibuprofen**															[207]
	**chlorhexidine**														Incl. *C. auris*	[208]
	**ganetespib**															[209]
	**HMA**														Incl. *Cryptococcus* sp.	[210]
**Natural Products**																
**Ess. Oil Extracts**	**thymol**															[211]
	**carvacrol**														Effective against biofilms of *C. auris*	[211,212]
	**acetophenone**															[213]
	**osthole**															[214]
	**houttuyfonate**															[215]
	**menthol**															[216]
	**tyrosol**														Effective against biofilms	[217]
	**allyl isothiocyan.**														Effective against biofilms	[218]
	**butylphthalide**														Effective against biofilms	[219]
	**glabridin**															[220]
	**chito-oligosacch.**															[221]
	**oridonin**															[222]
**Crude Ess. Oils**	**sea-buckthorn**															[223]
	**guava leaf**															[224]
	**frankincense**															[225]
	**TTO**															[226]
**Alkaloids**	**berberine**														Incl. *S. cerevisiae* and *T. marneffei*	[227,228,229,230,231,232]
	**palmatine**														Effective against biofilms	[233,234]
	**harmine**															[235]
**Other Terpenoids**	**guttiferone**															[236]
	**farnesol**														Effective against *C. auris* biofilms	[237]
	**asiatic acid**														Effective in vivo	[238]
**Other Phenols**	**magnolol**															[239]
	**diorcinol**														Extreme decrease in required dosage	[240]
	**proanthocyan.**															[241]
	**epigallocatechin**														Effective against biofilms	[242]
	**asarone**														Also clotrimazole	[243]
	**pyrogallol**															[244]
**Peptides**	**lactoferrin**														Other incl. *Cryptococcus* sp.	[201,245]
	**beauvericin**															[246]
**Novel Compounds**																
	**ATTAF-1 and -2**														Novel azole derivatives	[247]
	**31 and 42**														Novel azole derivatives	[248]
	**15 and 24**														Isoquinolone and phthalazinone deriv.	[249]
	**B-7b**														Novel berberine derivative	[250]
	**LQFM-79-81**														Novel guttiferone-A derivatives	[236]
	**phenylpentanol**														Novel phenylpentanol derivatives	[171]
	**AR-12**														Novel celecoxib derivative	[251]
	**SCY-078**														Glucan synthase inhibitor	[252]
	**DIBI**														Ion chelator	[253]
	**1 – 34c**														Novel caffeic acid derivative	[254]
	**chalcones**															[255]
	**AT406**														IAP Inhibitor, incl. *E. dermatitidis*	[256]
	**B2**														Piperidone derivative, incl. *C. neoformans*	[257]
	**H1-J10**														Novel HSP90/HDAC inhibitors	[258]
	**L1-C2**														Novel lipopeptides	[259]
	**AZD8055**														Novel TOR inhibitor	[260]
	**KEY:**			extremely strong synergy							synergistic in all strains tested					
				strong synergy							synergistic in >20% strains tested					
				weak synergy							synergistic in <20% strains tested					
				borderline synergy							not synergistic in any strains tested					

^1.^ “Incl.” refers to species included under “Other”.

## 2. Synergy between Azoles and Currently Available Antimicrobials

Existing antimicrobial pharmaceuticals that have been proven safe or tolerable for humans and have approval from regulatory bodies such as the FDA make an attractive starting point for antifungal synergy. Theoretically, combining antimicrobials might broaden their activity spectrum to include pathogens that are not susceptible to either drug as a monotherapy [261]. Although combining azoles with other classes of antifungals could be expected to often give synergy, certain antibacterial, antiparasitic and antiviral drugs have also been shown to interact synergistically.

### 2.1. Azole-Antifungal Synergy

Numerous antifungals, including azoles, allylamines and morpholines, target different points of the ergosterol biosynthesis pathway, as summarised in Figure 2. Given that the mechanisms of action of these drugs are often closely related, they are prime targets for investigation as potential azole synergists. Terbinafine is an allylamine that has been investigated across a broad array of azoles and various fungal and oomycete pathogens and was found to synergise with some systemic azoles (Table 2), particularly against azole-resistant *C. albicans* isolates. A combined treatment strategy using terbinafine and fluconazole was also shown to resolve persistent oropharyngeal thrush in a clinical setting [104,105,106]. For other species of *Candida*, however, terbinafine-azole synergy is weaker and only seen in azole-sensitive isolates, although the combination shifts inhibition from fungistatic to fungicidal [107]. Terbinafine and azoles have proven effective against clinical isolates of *Scedosporium prolificans*, a hard-to-treat pathogen of the lungs, sinuses and brain, at clinically achievable concentrations [108,109]. For other pathogens, terbinafine–azole synergy is narrow in spectrum, with only a very few isolates of *Aspergillus*, azole-resistant non-*albicans Candida* species, azole-resistant dermatophytes and *Pythium insidiosum*, (a fungus-like oomycete that is the cause of often-fatal pythiosis) affected [106,107,110,111,262,263].

Antifungals from the echinocandin class, which act by weakening the cell wall, were considered potentially attractive as azole synergists when they first became available; however, for most combinations, the pairs either synergise strongly but in a select subset of isolates, or work across a global spectrum of isolates but with only weakly synergistic interactions (Table 2) [113,114,115,116,117]. One particularly promising pair is micafungin and voriconazole, which strongly inhibits azole- and echinocandin-resistant *Candida auris*, bringing the required dosages of both to well within clinically achievable levels [115]. In a recent survey, anidulafungin, caspofungin and micafungin all displayed strong synergy with isavuconazole in *C. auris* [118]. Given the extreme level of resistance to azoles demonstrated by many *C. auris* isolates and the threat of emerging resistance to echinocandins, which are currently the front-line antifungal for *C. auris* infection, these combinations may warrant immediate clinical application [264,265,266].

Broadly speaking, most other market antifungals have demonstrated weak or no synergy with azoles. The topical polyene natamycin weakly synergises with voriconazole against a narrow range of azole-susceptible pathogens, while ciclopirox synergises strongly with itraconazole but only in a small number of dermatophytes [119,120]. Flucytosine and even voriconazole have been tested with other azoles but were indifferent or only weakly synergistic for various fungal species, including *C. auris* and pathogens in the order Mucorales [121,122]. Amorolfine, however, which acts on the ergosterol biosynthesis pathway subsequent to azoles (Figure 2) displayed more promise, synergising with systemic and topical azoles against dermatophytes in vitro, and demonstrating efficacy in an open randomised clinical trial of notoriously refractive onychomycosis [123,124,125]. While promising, this combined treatment strategy for topical fungal infections is yet to make it to market.

A few recently developed novel antifungals that are not yet available commercially appear promising as azole synergists. K20, a novel fungus-specific amphiphilic aminoglycoside that inhibits *Fusarium* spp. and a variety of yeast pathogens, displayed strong synergy with a wide variety of systemic azoles for almost all *Candida* isolates tested [127,267]. Oxadiazole-tagged macrocyclic peptides, which are capable of inhibiting azole-resistant *C. glabrata* and *C. tropicalis* strains also interacted synergistically with fluconazole, but with less synergy and a narrower spectrum of activity than K20 (Table 2) [128].

**Figure 2 pharmaceuticals-15-00482-f002:**
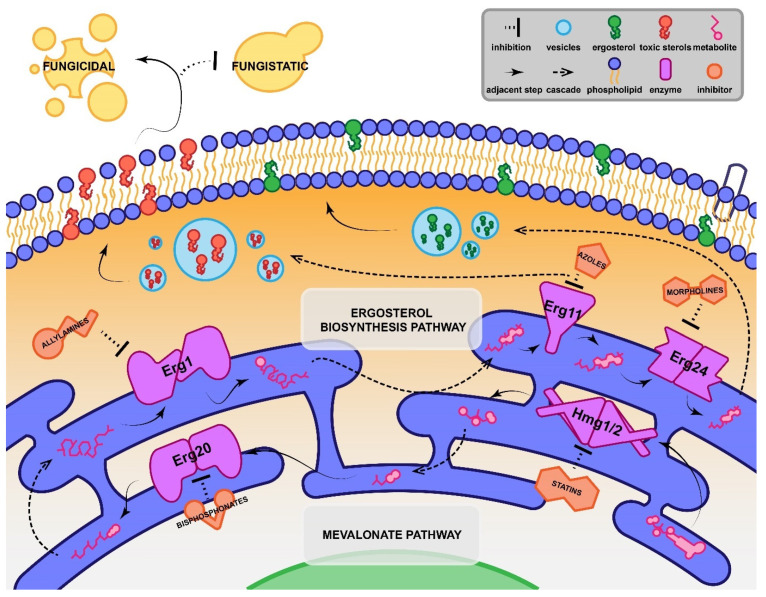
Proposed mechanism of synergy between azoles and inhibitors that operate on the ergosterol biosynthesis and mevalonate pathways. In fungi, the synthesis of ergosterol occurs primarily in the endoplasmic reticulum, with the final product packaged into vesicles to be incorporated into the membrane [268]. Azole drugs inhibit Erg11, preventing lanosterol from being converted into dimethyltrienol and leading to the build-up of toxic methyl sterols. These are incorporated into the membrane instead of mature ergosterol, causing a loss of membrane structure and an inability to divide and resulting in the fungistatic arrest of growth [47]. Synergistic inhibitors co-operate at points up- and down-stream of Erg11, increasing the generation of toxic ergosterol precursors and other terpene-derived metabolites. The mevalonate pathway, upstream from the ergosterol biosynthesis pathway, is responsible for the biosynthesis of squalene, a precursor to all fungal membrane sterols. Statins like atorvastatin inhibit the HMG-CoA reductases Hmg1 and Hmg2, which are responsible for the production of mevalonate from HMG-CoA [269]. Further downstream, bisphosphonates like zoledronate inhibit farnesyl pyrophosphate synthase, or Erg20, which catalyses the production of farnesyl pyrophosphate from dimethylallyl pyrophosphate [270]. In the ergosterol biosynthesis pathway, squalene is converted into squalene epoxide by Erg1, a squalene epoxidase, which can be inhibited by allylamines like terbinafine [271]. Downstream from Erg11, dimethyltrienol is converted again into dimethylzymosterol by Erg24, a sterol reductase that is inhibited by morpholine antifungals such as amorolfine [47,272]. The resulting destabilisation of the cell membrane means synergy can often produce a fungicidal effect in the pathogen.

### 2.2. Azole–Antibacterial Synergy

Among the antibacterials, the sulfa-based drugs have shown the greatest potential to date for azole synergy (Table 2). These inhibit folic acid biosynthesis and several, including the widely available antibacterial sulfamethoxazole, have displayed strong synergy with azoles in most azole-resistant *Candida* isolates, including isolates of *C. auris* [129,130]. Tetracycline-based antibacterial agents have demonstrated synergy with azoles specifically against azole-resistant pathogens (Table 2). Doxycycline, tigecycline and minocycline could all potentially be used to improve therapy with azoles for persistent cases of candidiasis, aspergillosis and fusariosis [131,132,133,134,135,136,137]. Minocycline in particular has recently demonstrated potential as an itraconazole synergist in *C. neoformans* and *Scedosporium* sp. [138,139]. This synergy has been consistently reproduced in in vivo infection models and against sessile forms of *Candida*, with some demonstrating efficacy against biofilms. Colistin is a last-resort antibiotic that has recently displayed promising synergy in vitro with isavuconazole in *Aspergillus* spp. and *Candida auris,* and in vivo in *Candida albicans* [93,96,143,144,252,253].

Other antibacterial compounds have failed to display any real potential as combined antifungal therapies. Gentamicin was synergistic with fluconazole in biofilms produced by some resistant *Candida* species. Linezolid failed to produce true synergy with any azoles tested but it did reduce the required dosage of both drugs to a clinically achievable level for a limited spectrum of fungi (Table 2) [140,141]. While reducing the dose is one of the main goals when developing novel combination treatments, in the wake of more viable therapeutic leads it seems unlikely that drugs like linezolid will see further development. Furthermore, the in vivo use of voriconazole and clarithromycin, another antibacterial found to be synergistic in vitro, was found to cause acute kidney damage, illustrating the potential for undesirable consequences when exploiting antimicrobial synergy [273].

### 2.3. Azole–Antiparasitic Synergy

Only a limited number of antiparasitic compounds have been explored for azole synergy in recent years, but some of them show promise. Chloroquine and artemisinin interacted synergistically with azoles against azole-resistant *Candida* strains, while pyrvinium pamoate–azole synergy was observed in a broad suite of dermatophytes [145,146,147,148]. Mefloquine and related compounds displayed limited synergy in *C. neoformans*, but did potentiate a strong fungicidal effect when combined with fungistatic fluconazole [149].

### 2.4. Azole–Antiviral Synergy

Antivirals have recently been investigated as potential antifungal synergists, with the anti-retrovirals showing the most promise. Saquinavir and ritonavir were effective at synergistically cooperating with azoles against *Histoplasma*, a systemic fungal pathogen [150]. Other antivirals like ribavirin and 2-adamantanine have shown promise at treating biofilms of azole-resistant *C. albicans* and potentiating the antifungal activity of azoles from fungistatic to fungicidal, respectively [151,152]. Lopinavir is an antiviral that shows extremely strong synergy with voriconazole in a majority of azole-resistant *C. auris* strains, and certainly warrants further investigation [153].

## 3. Active Efflux Modulators

The active, ATP-dependent transport of antimicrobial compounds out of the fungal cell is one of the most concerning mechanisms of antifungal resistance emerging today. It is the principal mechanism of azole tolerance in *C. glabrata* and in a significant portion of *C. auris* and azole-resistant *C. albicans* isolates [44,274,275]. There is also evidence that active efflux is responsible for azole resistance in some *Aspergillus* species and dermatophytes [276,277]. Due to the importance of pumps for resistance in bacterial pathogens and malarial parasites, inhibition of active efflux is a popular target in antimicrobial discovery and the development of combined antifungal treatments [278,279].

Most efflux pump inhibitors that have been investigated as azole synergists interact directly with the membrane-bound pump. Several drugs that affect the intracellular homeostasis of calcium by blocking calcium channels have been found to have some affinity for membrane-bound transporters, particularly tetrandrine and verapamil. Traditionally used as immunomodulators and vasodilators, these have produced antifungal synergy with a variety of azoles. Tetrandrine combined with posaconazole has been proposed as an effective solution for persistent and temporary candidiasis [154,155,156,157,158,159,280]. Eucalyptal D and dodecenoic acid are essential oil extracts that have been shown to directly antagonise ABC transporters [160,161]. These exhibited strong synergy with fluconazole and itraconazole, respectively, in azole-resistant *Candida.* Azoffluxin, an oxindole Cdr1 inhibitor, has been shown to synergise strongly with fluconazole in all non-clade III *C. auris* and azole-resistant *C. albicans* strains, both in vitro and in vivo [162]. Other receptor antagonists may be cross-reactive to membrane-bound transporters; for example, ospemifeme is a promising therapeutic lead that has a broad spectrum of activity and synergises very potently with fluconazole (Figure 3g) [159,163].

Several efflux inhibitors that show promising synergy with azoles do not directly bind pump proteins but interfere with efflux through other mechanisms. Palmarumycin P3 and phialocephalarin B are naturally occurring quinone derivatives that synergise with fluconazole in pump-dependent, azole-resistant *C. albicans*. It appears that the mechanism of synergy is due to the ability of these derivatives to directly inhibit nuclear transcription factors, modulating the expression of the principal pump-coding gene *MDR1* [164]. Geraniol is a unique synergist that causes the localisation of pumps to become dysregulated, preventing them from being incorporated into the membrane and resulting in weak synergy [165]. The controversial anti-cancer drug ponatinib is able to inhibit active efflux in multiple yeast pathogens by interfering with the proton motive force, producing strong synergy with fluconazole in all strains tested [196]. Other potential co-drugs like cationic triphenylphosphonium have been shown to improve the activity of efflux pump inhibitors, suggesting the potential development of a triple-drug antifungal treatment strategy [281].

Many other compounds currently being investigated for their ability to synergise with azoles promote the downstream inhibition of efflux, despite no direct action on the pumps themselves or their expression. Haloperidol and promethazine are repurposed drugs that have displayed the ability to modulate active efflux. Both show great promise as azole adjuvants for the treatment of cutaneous mycoses and dandruff [18,173,174]. Thymol and carvacrol are terpenoids extracted from thyme leaves that interact synergistically with azoles and inhibit efflux in azole-resistant *Candida,* including *C. auris* [211,212,282]. Numerous other natural products with synergistic potential have displayed similar indirect anti-efflux properties [203,211,223,227,233].

It should be noted that fungal drug efflux pumps have an extremely broad spectrum of substrates that they transport [283]. Given this, many of the compounds discussed in this review may be substrates of efflux pumps that compete with azoles, preventing the transport of the azole and prolonging its effect. This competition may contribute to the synergy observed between some drug pairs where the underlying mechanism is currently unknown.

## 4. Repurposing Other Pharmaceuticals

Repurposing existing drugs can short-cut the process of drug development and regulatory approval. In recent years, a popular approach in drug discovery has been to screen libraries of existing compounds for novel repurposed uses, including antifungal activity. The resulting “hits” may have completely different mechanisms of action to any other currently available antifungal, opening new avenues for drug design. Many of these repurposed antimicrobials are also tested for their capacity as antifungal adjuvants [284,285].

### 4.1. Statins

Statins are common anti-cholesterol medications and until recently were considered one of the most promising routes for antifungal discovery. Statins operate on HMG-CoA reductase in the mevalonate pathway, upstream from azole-targeted demethylases (Figure 2) [269]. Unfortunately, however, as demonstrated in Table 2, synergy between azoles and statins is often minor or bordering on indifferent. In addition, while it might appear that some statin–azole pairings have a decent spectrum of activity, most studies have tested only 1–3 strains per species. Their applicability to a wide variety of mycoses is therefore difficult to gauge [166,167,168,169,170]. Where synergy has been observed, the mechanism was shown to be primarily driven by the co-operation of the drug pairs on the same pathway (Figure 2) [286]. Newer statins such as pitavastatin have been found to exhibit extremely strong antifungal synergy with fluconazole in azole-resistant *Candida*, which may reignite future interest [171]. Statins can unfortunately have adverse off-target effects and drug interactions that range from unpleasant to lethal, especially for already vulnerable mycosis patients. These include diabetes, liver cirrhosis, irreparable damage to skeletal muscle and sexual dysfunction [287,288].

### 4.2. Bisphosphonates

Bisphosphonates are anti-osteoporosis drugs and show promising synergy with fluconazole. Like statins, bisphosphonates operate on the mevalonate pathway where they target farnesyl pyrophosphate synthase (or Erg11; Figure 2) [289]. Of the bisphosphonates tested, zoledronate resulted in strong synergy across numerous strains of *Cryptococcus* tested in vitro and in an in vivo model and significantly limited the development of antifungal resistance [172]. Due to their propensity to bind to bone mineral and their implication in osteonecrosis, market bisphosphonates have limited applicability for the treatment of invasive mycoses [289,290]; however, their antifungal synergy and potent immunostimulatory properties make them attractive lead compounds for further development [291].

### 4.3. Repurposing Miscellaneous Pharmaceuticals

Although most fail to produce strong, broad-spectrum synergy and may have undesirable anti-inflammatory or immunosuppressive effects, certain immunomodulators could be useful as lead compounds for development as synergists to treat *Candida* biofilms [175,176,177,178,292]. The antihistamine promethazine appears potentially attractive as a novel topical anti-dandruff and anti-tinea treatment, as it synergised strongly with azoles in all strains of dermatophytes tested [18,173,174].

Some psychoactive drugs have displayed limited synergy with azoles along a narrow spectrum of activity. Bromperidol and fluoxetine show limited potential as azole adjuvants for the treatment of candidiasis, while the commonly prescribed antidepressant sertraline synergised weakly with azoles in the opportunistic yeast *Trichosporon* [179,180,181]. Haloperidol is an antipsychotic that may be more promising as a topical combined treatment due to its strong synergy with fluconazole and itraconazole in many strains of dermatophytes [18,173].

Two of the most attractive groups of compounds for developing new synergists are calcineurin inhibitors and calcium channel blockers. These drugs modulate calcium ion homeostasis, which is vital for cellular signalling, and have been considered promising antifungal leads for a variety of diverse pathogens for more than a decade [184,185,186,293]. Calcium channel blockers have also been shown to further enhance the synergy between fluconazole and doxycycline in a series of three-way checkerboards [131]. The calcium channel blockers tetrandrine and verapamil are discussed above in the context of their role in efflux, but their ability to disturb calcium homeostasis is likely also part of their antifungal effect. Other inhibitors like cyclosporine and tacrolimus target calcineurin and calmodulin, a calcium-activated complex responsible for the up-regulation of genes related to growth and the fungal stress response (Figure 3c) [18,182,183,184,185,187,294]. Tacrolimus specifically produces significant synergy for the majority of dermatophyte species, while cyclosporine reliably and potently interacts with fluconazole in *C. albicans* [183,188,189]. It should also be noted that calcineurin inhibitors like tacrolimus directly affect the ATPase activity of efflux pumps in addition to acting on calcineurin, thereby both directly and indirectly inhibiting active efflux [295].

The growing drug repurposing initiative has yielded synergists with novel mechanisms of action with significant therapeutic potential, as illustrated in Figure 3. Fungal membrane proton pumps are vital enzymes, generating the membrane potential required for membrane-bound transporter function and the uptake of nutrients required for ATP synthesis, thereby enabling ATP-dependent drug efflux [52]. A wide variety of proton pump inhibitors have been shown to produce strong synergy with fluconazole in azole-resistant *Candida* isolates [194]. HSP90 (HSP82 in yeast) inhibitors like geldanamycin and ganetespib repress the fungal response to stress, reducing the survival of yeasts during azole-induced inhibition (Figure 3a) [195,197,209]. Histone deacetylase inhibitors have been popular in the development of antifungal therapies but surprisingly few have been tested for synergy with azoles. One exception is givinostat, a potential anti-aspergillosis treatment when paired with posaconazole [198]. Lonafarnib inhibits farnesyltransferase, preventing vital post-translational modifications of fungal proteins and producing moderate azole synergy in *Aspergillus* (Figure 3b) [199]. Other recently discovered azole synergists with more limited utility inhibit superoxide dismutase, chemosensitising *C. albicans* to oxidative stressors induced by azole treatment (Figure 3f) [200]. Others such as DIBI, lactoferrin, D-penicillamine and EDTA chelate ions vital for enzymatic function, resulting in dysregulated apoptosis in the cell, but these result in only a weak or narrow-spectrum antifungal synergy (Figure 3d) [201,202,245,253].

Some repurposed pharmaceuticals interact synergistically with azoles through a mechanism that has yet to be fully elucidated. Licofelone failed to make it through clinical trials as an anti-arthritic but was shown to abrogate azole resistance in biofilms of *C. albicans* [203]. Phenylbutyrate is an aromatic fatty acid used to treat hyperammonaemia and has been shown to weakly synergise with various systemic azoles against resistant *Candida* species [204]. Other repurposed inhibitors including sedatives, antiseptics, diuretics and analgesics have displayed some promising synergy with azoles against a narrow spectrum of strains that may warrant further exploration [205,206,207,208,210].

**Figure 3 pharmaceuticals-15-00482-f003:**
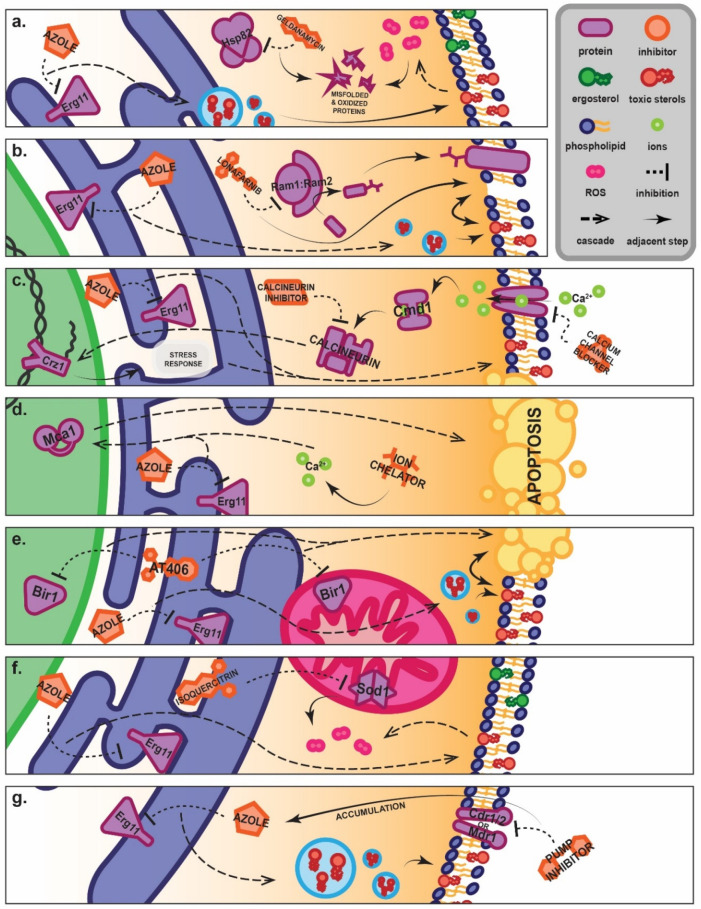
**Proposed novel mechanisms of synergy between azoles and inhibitors that operate on entirely separate pathways.** (**a**) HSP82 inhibitors like geldanamycin prevent the association of Hsp82 with proteins, inhibiting proper folding of nascent proteins and degradation of senescent proteins. Accumulation of toxic oxygen radicals results in oxidation of proteins, which would ordinarily be degraded. HSP82 inhibitor–azole synergy therefore appears to rise from the accumulation of oxidatively damaged, toxic proteins [195]. (**b**) Inhibition of protein farnesylation by farnesyltransferase (Ram1:Ram2) inhibitors such as lonafarnib results in reduced translocation of membrane-bound proteins. This decline in the population of membrane proteins combines synergistically with the azole-induced build-up of toxic sterols, resulting in increased membrane instability [199]. (**c**) Calcium channel blockers and calcineurin inhibitors prevent the activation of the calcineurin complex by calmodulin (Cmd1). This results in an inability of calcineurin to dephosphorylate Crz1, which would ordinarily mobilise it to the nucleus. Crz1 is a transcription factor responsible for the regulation of several stress-related genes. Calcium channel blockers and calcineurin inhibitors therefore impair the cellular stress response, sensitising the cell to the antifungal effect of azoles [182,280,296]. (**d**) Ion chelators like DIBI and D-penicillamine bind to ions and disrupt cellular ion homeostasis. Evidence suggests that it is the disturbance of calcium homeostasis that results in the promotion of metacaspase (Mca1)-dependent apoptosis when paired with azoles, synergistically enhancing the fungicidal effect [202,253]. (**e**) AT406 is an antagonist of the inhibition of apoptosis proteins (IAPs) such as Bir1, which is present in both the mitochondrion and the nucleus. There is evidence that membrane weakness due to toxic sterol build-up improves the pro-apoptotic effects of AT406 [256]. (**f**) Some novel synergists, such as isoquercitrin, have demonstrated the ability to inhibit mitochondrial superoxide dismutase, Sod1. Sod1 becomes unable to neutralise harmful reactive oxygen species that accumulate during azole treatment, resulting in rapid accumulation of radicals and potentiating a toxic oxidative effect [200]. (**g**) Direct and indirect inhibitors of both ABC transporters such as Cdr1 and MFS transporters such as Mdr1 prevent the active efflux of toxic compounds such as fluconazole out of the cell, resulting in an accumulation of the drug and extending its antifungal effect. In turn, the destabilised membrane may reduce or prevent incorporation of transmembrane proteins including pumps, further reducing the efflux capabilities of the cell [173,183,240].

## 5. Azole Synergy with Natural Products

Nature has long been the source of novel compounds, including important anti-cancer and antimicrobial chemotherapies [297]. Table 3 lists naturally produced synergists that have been included in this review alongside their original biological source.

Some essential oils and essential oil extracts have shown strong potential as lead compounds, but most are only effective at concentrations that are not clinically achievable or are active only in particular isolates. Thymol and carvacrol are well-characterised terpenoids from thyme and oregano essential oils (Table 3) and have been found to synergise with fluconazole in wild-type *C. albicans* and have displayed some activity in *C. auris* biofilms [211,212,282]. Acetophenone is a small ketone present in many foods that showed promise as a topical antifungal when paired with ketoconazole [213]. Osthole from tonka bean oil and houttuyfonate from fish mint oil both displayed excellent synergy with azoles in azole-resistant species of *Candida* [214,215]. Oridonin is a staple of traditional Chinese medicine that has displayed strong synergy with common azoles in resistant isolates of *Candida* [222]. Menthol extracted from mint synergised well with itraconazole, but only for a fraction of the *Candida* strains tested [216]. Glabridin from liquorice root synergised with voriconazole in all *A. fumigatus* strains tested [220]. Several oil extracts displayed excellent anti-biofilm activity when combined with fluconazole, including chito-oligosaccharides, tyrosol from olive oil, allyl isothiocyanate from mustard oil and butylphthalide from celery oil [217,218,219,221]. Some crude essential oils have also been investigated for potential synergy, including oils from Indian frankincense, tea tree, sea buckthorn and guava leaf. While some promising anti-*Candida* and anti-dermatophyte synergy was observed, these crude oils are too complex to be called therapeutic leads, and more refined bioactive fractions need to be identified [223,224,225,226].

Alkaloid and terpenoid metabolites make up many of the most promising naturally derived synergists. The alkaloid phytochemicals berberine and palmatine are particularly attractive, displaying synergy with fluconazole in both planktonic and biofilm forms of nearly all strains tested in the *Candida* genus, *Saccharomyces cerevisiae* and *Talaromyces marneffei* [227,228,229,230,231,232,233,234]. Harmine is an alkaloid that interacts extremely synergistically with multiple systemic azoles, but only in a fraction of the strains tested, while guttiferone, a terpenoid, synergises less acutely but with more total strain coverage, particularly in non-*albicans* species of *Candida* [235,236]. Berberine and guttiferone are two of the very few phytochemicals actually taken past the point of a therapeutic lead, undergoing chemical modifications and development into more synergistic novel derivatives in further studies [236,250,298]. Farnesol is an isoprenoid that displays limited or no synergy in planktonic forms of *C. auris* but is highly synergistic against its biofilms. This contrasts with asiatic acid, a terpenoid that synergises with fluconazole only in planktonic *Candida* or in vivo, but not in biofilms [238]. Farnesol may, thus, be a promising tool for fighting highly problematic biofilms of *C. auris* [237].

Phenol derivatives are an important class of organic phytochemicals, often vital for mediating the plant response to stress [299]. Many phenolics have been shown to have antimicrobial activity, and several may be potential synergistic azole co-drugs [300]. Both the lignan magnolol and the diphenol diorcinol have proven effective against every *Candida* strain tested, with the former synergising well with fluconazole and the latter not technically synergising but sharply decreasing the required inhibitory dosage, bringing it to well within a clinically achievable concentration [239,240]. In contrast, proanthocyanidin, a plant polyphenol, synergised with fluconazole very weakly and in only a small number of the azole-resistant non-*albicans Candida* isolates tested [241]. Other phenolics show greater promise, particularly for the treatment of cutaneous or oropharyngeal candidiasis. Epigallocatechin gallate and asarones both synergise with topical azoles, with the former particularly effective against *Candida* biofilms [242,243]. Pyrogallol is a phenol that synergises with various market antifungals by inhibiting active efflux in both azole-susceptible and azole-resistant *Candida* [244]. Catechol, while unable to inhibit *Candida* itself, potentiates the antifungal activity of azoles and polyenes, and prevents up-regulation of virulence-associated genes. Curiously, catechol did not reduce the viability of *Candida* biofilms, but did reduce their hydrophobicity [301].

Antimicrobial peptides are found in cells from all taxonomic Kingdoms and are vital for the defence against infection, potentially synergising with other antimicrobials [302]. Synergy has been seen with the milk protein lactoferrin with azoles in resistant isolates of *Candida,* but not *Cryptococcus* [201,245]. Beauvericin is an antibiotic and insecticidal peptide derivative called a depsipeptide that synergises well with fluconazole. With only a limited group of azole-susceptible strains of *C. albicans* tested, however, it remains unclear whether it can be called a truly promising therapeutic lead [246].

## 6. Azole Synergy with Novel Compounds

Numerous new compounds that have been shown to have good antifungal activity as a monotherapy have also been found to synergize with azoles. Synergy is often inconsistent for azole-susceptible and azole-resistant strains, however. Consistent with the cross-resistance observed between azoles, novel azole derivatives are generally effective only in sensitive yeasts and not in resistant ones [247,248]. Novel azoles conjugated with triphenylphosphonium cations have displayed improved mitochondrial targeting and have shown an improved fungicidal effect when combined with Hsp90 inhibitors [303].

Several of the more promising novel compounds currently under investigation are chemically modified derivatives of promising therapeutic leads. Derivatives of the aforementioned isoquinolone and phthalazine, natural metabolites berberine, piperidol, caffeic acid and guttiferone, the anti-inflammatory celecoxib, and phenylpentanol all demonstrated strong synergy with fluconazole in a significant portion of *Candida* strains tested, and in particular in azole-resistant strains [171,236,249,250,251,254,257]. Other promising novel compounds like beta-glucan synthase inhibitor SCY-078, ion chelator DIBI, TOR inhibitor AZD8055, efflux modulators, and a group of novel antifungal chalcones have all proven highly synergistic against azole-resistant *C. albicans*, *C. glabrata* and other *Candida* species with reduced sensitivity to azoles [173,252,253,255,260]. A group of novel ultra-short cationic lipopeptides were not able to fully synergise with fluconazole, but did interact additively [259].

Two promising classes of compounds that have been developed in the past year are “dual-inhibitors”, which are single compounds that are designed to attack more than one druggable target at once. One group of these that was strongly antifungal contains a piperazine moiety, allowing it to inhibit 14α-demethylase, and a zinc binding group to inhibit HDAC [304]. Another used fluconazole conjugated with COX inhibitors and was able to consistently inhibit pathogenic *Candida* [305]. As these dual inhibitors are single compounds, they cannot be considered truly synergistic; however, a class of novel Hsp90/HDAC dual inhibitors has displayed strong synergy with fluconazole in azole-resistant *Candida* [258].

An interesting novel antifungal mechanism of action is the promotion of dysregulated apoptosis in the fungal cell. Control of apoptosis in yeasts is partially governed by the regulation of Inhibitors of Apoptosis Proteins (IAPs), which prevent progression to cell death [306]. A new IAP antagonist, AT406, promotes apoptosis in *C. albicans* and the outbreak pathogen *Exophiala dermatitidis*, sensitising the cell to the oxidative stress produced by the azoles (Figure 3e) [256]. This is a truly novel mode of fungicidal action that may be a source of an entirely new class of azole synergists.

## 7. Conclusions

With the rise of opportunistic and emerging fungal pathogens and increasing rates of antifungal resistance, there is an urgent need for new therapies in our antifungal toolbox. As this review has shown, a reliable path to success is to improve commonly used azole antifungals with synergists, and there is substantial diversity in the compounds and approaches that have yielded synergy. Although high-throughput screening has become a popular method for discovering new therapeutic agents, a rational, target-based approach to drug discovery may yield more reliable and effective therapeutic leads. Compounds co-operating with azoles on the ergosterol and mevalonate biosynthesis pathways have displayed consistent synergy in various fungal pathogens. Rational drug design, building on a known mechanism of action or starting with already approved drugs with known pharmacokinetic data, may take newly developed drugs into market more rapidly. On the other hand, hypothesis-free drug screening initiatives may yield novel synergies that would otherwise be undiscovered, opening new avenues for drug design.

There are several gaps in current studies that await further research. From Table 2, there is a clear focus on developing combined treatment strategies to combat *Candida*, particularly azole-resistant clinical isolates, and with a few notable exceptions there is a paucity of data exploring azole synergy in *Cryptococcus* and filamentous fungi. There is also a focus on improving the systemic azoles, particularly fluconazole, while for topical pathogens where oral bioavailability is not required, more could be gained by exploring other azoles. Finally, any translation of synergy into clinical use must deal with the issue of co-administration of two (or more) compounds. New systems that package drugs into nanoparticle delivery systems or co-crystallise compounds into a single formulation, may enable the development of single-dose synergistic treatments [307,308]. There are currently no azole-based antifungal combinations used to treat mycoses, but the need for new treatments and the threats to azoles from intrinsic and acquired resistance make drug synergy an increasingly attractive avenue for antifungal development.

## Figures and Tables

**Table 3 pharmaceuticals-15-00482-t003:** Bioactive natural products named in this review and their original biological source(s).

		Source		
Type	Synergist	Common Name	Latin Name	Ref.
**Essential Oil Extracts**	thymol	thyme, ajwain, wild bergamot	*Thymus vulgaris, Trachyspermum ammi, Monarda fistulosa*	[211]
	carvacrol	oregano, thyme, marjoram	*Origanum vulgare, Thymus vulgaris, Origanum majorana*	[211]
	acetophenone	apple, apricot, beef, cheese, croton	*Malus domestica, Prunus armeniaca, Bos taurus, Croton* sp.	[213]
	osthole	snowparsley, wild celery, shishiudo	*Cnidium monnieri, Angelica archangelica, Angelica pubescens*	[214]
	houttuyfonate	fish mint	*Houttuynia cordata*	[215]
	menthol	wild mint, peppermint	*Mentha arvensis, Mentha piperita*	[216]
	tyrosol	olive, argan	*Olea europaea, Argania spinosa*	[217]
	allyl isothiocyanate	mustard, radish, horseradish, wasabi	*Sinapis alba, Raphanus raphanistrum, Armoracia rusticana*	[218]
	butylphthalide	celery	*Apium graveolens*	[219]
	glabridin	liquorice	*Glycyrrhiza glabra*	[220]
	oridonin	blushred	*Rabdosia rubescens*	[222]
**Crude Essential Oils**	sea-buckthorn	sea-buckthorn	*Hippophae rhamnoides*	[223]
	guava leaf	guava, pineapple guava	*Psidium guajava, Acca sellowiana*	[224]
	frankincense	Indian frankincense	*Boswellia serrata*	[225]
	TTO	tea tree	*Melaleuca alternifolia*	[226]
**Alkaloids**	berberine	barberry, tree turmeric, prickly poppy	*Berberis vulgaris, Berberis aristate, Argemone mexicana*	[228,229,230,231,232]
	palmatine	Amur cork tree, yanhusuo	*Phellodendron amurense, Corydalis yanhusuo*	[233,234]
	harmine	wild rue, ayahuasca	*Peganum harmala, Banisteriopsis caapi*	[235]
**Other** **terpenoids**	guttiferone	boarwood root	*Symphonia globulifera*	[236]
	farnesol	plants, animals and fungi		[237]
**Other phenols**	magnolol	Chinese magnolia, southern magnolia	*Magnolia officinalis, Magnolia grandiflora*	[239]
	diorcinol	fungal symbiont	*Epichloe bromicola*	[240]
	proanthocyanidin	pine bark, cranberries, grape seeds	*Pinus* sp., *Vaccinium oxycoccus*, *Vitis labrusca*	[241]
	epigallocatechin	black tea, white tea, green tea	*Camellia sinensis*	[242]
	asarone	sweet flag, wild ginger	*Acorus calamus, Asarum* sp.	[243]
**Peptides**	lactoferrin	bovine and human (milk, mucous)	*Bos taurus, Homo sapiens*	[201,245]
	beauvericin	white muscardine, *Fusarium*	*Beauveria bassiana, Fusarium* sp.	[246]

## Data Availability

Data sharing not applicable.

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
