# Peer review of "Augmenting Azoles with Drug Synergy to Expand the Antifungal Toolbox"

_pharmaceuticals, 2022, doi:10.3390/ph15040482_

Round 1

Reviewer 1 Report

This is a well written and improved version of a previous submission, with excellent easily read and comprehensive content. It  includes very clear written descriptions of some complex interactions in figure legends. The review will be of value as it becomes increasingly important to select methods that will extend the value of the azole drugs by using drug synergy.

The MS could be significantly improved by:

1. Increasing the size of the text in Table 1 (such as in table 3) and preferably keeping the table to a single page.

2. Increasing the size of the text used in Table 2 and placing headings and new pages of this large table.

3. The text across all the figures indicating embedded objects or compounds and in inserted keys is too small (the text used in Figure 3 is of a more appropriate and readable size).

4. Line 113. It might be useful to reference the contribution of Lepesheva's group toward the discovery of the tetrazole VT-1596 and the review of Monk and Keniya J Fungi (Basel). 2021 Jan 20;7(2):67. doi: 10.3390/jof7020067 on structure based discovery of drugs targeting sterol 14alpha-demethylases.

5. Recent work that could be usefully discussed involvesto the indirect inhibitor of Pma1 ponatinib which synergizes with FLC. Lin Liu et al 2021 https://doi.org/10.1111/1751-7915.13814.

Author Response

1. Increasing the size of the text in Table 1 (such as in table 3) and preferably keeping the table to a single page.

Response:  We have increased the text size slightly but unfortunately we can’t do more without ruining the table’s formatting and legibility, and we need to work within the constraints of the template. We are leaving it to the publisher to determine if the current table is acceptable.

2. Increasing the size of the text used in Table 2 and placing headings and new pages of this large table.

Response: As above we can’t increase the font size in the template without ruining the table, and the template does not seem to allow adding headings on new pages or changing the placement of the table to a different place in the manuscript. We are leaving it to the publisher to determine if the current table is acceptable.

3.  The text across all the figures indicating embedded objects or compounds and in inserted keys is too small (the text used in Figure 3 is of a more appropriate and readable size).

Response:  We have slightly increased the size of the text in Figs. 1 and 2. The text in the figures is legible when printed on A4 paper and will be further improved when the high-resolution figures are embedded.

4.  Line 113. It might be useful to reference the contribution of Lepesheva's group toward the discovery of the tetrazole VT-1596 and the review of Monk and Keniya J Fungi (Basel). 2021 Jan 20;7(2):67. doi: 10.3390/jof7020067 on structure based discovery of drugs targeting sterol 14alpha-demethylases.

Response:  We have added these references to the review (Line 72).

5.  Recent work that could be usefully discussed involves to the indirect inhibitor of Pma1 ponatinib which synergizes with FLC. Lin Liu et al 2021 https://doi.org/10.1111/1751-7915.13814.

Response:  We have added a comment on this very relevant study (Lines 317-319). We thank the reviewer for bringing it to our attention.

Reviewer 2 Report

Dear authors,

Thank you for this revised version of the manuscript. Congratulations for the excellent review! I have no further comments to add.

Best regards

Author Response

We thank the reviewer for their kind comments.

Reviewer 3 Report

Very good review. I recommend acceptance in the present version.

No particular comments.

Author Response

We thank the reviewer for their kind comments.

This manuscript is a resubmission of an earlier submission. The following is a list of the peer review reports and author responses from that submission.

Round 1

Reviewer 1 Report

The MS provides a useful description and insight into the concept of synergy due to compounds that substantially increase azole susceptibility, thereby minimizing opportunity for the the acquisition of azole resistance and potentially circumventing the impact of azole resistance when it has occurred. This approach has yet to be made commercially commercially but could become an important practice, particularly with the acquisition of resistance to the azole drugs in organisms such as Candida albicans and Candida parapsilosis  and filamentous fungi such as Aspergillus fumigatus, the emergence of intrinsically resistant organisms such Candida glabrata,  Candida auris and the mucormycetes, plus the difficulties with the long term treatments required for Cryptoccocus neoformans infections. However the review has some significant limitations that should be addressed.

  1. Considering the importance of antifungal resistance in the context of this review, the responsible mechanisms are not adequately described/spelled out - i.e. target-based resistance involving mutations or intrinsic substitutions  in the CYP51 gene, overexpression of Cyp51  and drug efflux pumps due to aneupoidy,  azole interactions with transcription factors affecting CYP51 and efflux pump expression, acquired mutations in transcription factors  that render efflux pump expression constitutive, and mutations that render Erg3 ineffective and thereby prevent toxicity due to misdirected metabolism of lanosterol.
  2. How azoles target sterol 14∝-demethylases is addressed, but the conclusion that there iemains limited potential for developing azole antifungals is a little premature. Noting that medicinal chemistry offers limited opportunity for azole development neglects the opportunity for structure-directed antifungal discovery based on the large number of crystal structures of fungal lanosterol 14∝-demethylase in complex with a wide range of azole drugs and agrochemicals now available in the Protein Data bank.
  3.   The description of the different generations of azole drugs is incomplete/incorrect and does not reflect ongoing innovation with these compounds. The first generation of azole drugs were the imidazoles (e.g ketoconazole). The triazoles fluconazole and itraconazole are second generation azole drugs (first generation triazoles) while voriconazole and posaconazole are third generation azole (second generation triazoles) drugs. The prodrug isavuconazolium could be considered as a fourth generation azole while the tetrazoles VT-1161 and VT-1129 in clinical development could be regarded as fifth generation azoles. While there remains high levels of mortality (e.g. 30-50% for C. albicans ) among individuals with invasive/disseminated fungal infection, additional problems include the fact that such individuals have weak or disabled immune systems and their fungal infections are often diagnosed too late. The statement that "advances in combating opportunistic fungal infections have primarily been in the antiviral space" is an overreach. The MS reflects developments associated HART therapy against HIV-AIDs in the 1990s but the development of effective antivirals since then have been less than impressive. It should be noted that all generations of azole drugs are susceptible to drug efflux by both ABC and MFS transporters. In addition, the discovery and focus on synergistic compounds that convert the fungistatic azole drugs into fungicides would be especially important. Finally, it is particularly important in the context of this review to define/refer to the criteria used to define compounds as azole antagonists, having additive activity with azoles or being synergistic with these drugs.
  4. It is stated in line 138 that the review focuses on developments with synergistic compounds identified in the past 5 years. It would be a good idea to include for the reader references to earlier research/reviews that have considered aspects of this area e.g. Targeting efflux pumps to overcome antifungal drug resistance. Holmes AR, et al doi: 10.4155/fmc-2016-0050.)
  5. Table 2 could be improved by  defining the azole abbreviations, placing the heading at the top of each page and by indicating all species tested. There should be a differentiation between <10% of strains affected and no strains affected.
  6. Lines 151-152 are poorly expressed. Suggested improvement - "Although the combination of an azole with other classes of antifungal agent could be expected to give synergy, certain antibacterial, antiparasitic and even antiviral drugs have been shown to act synergistically."
  7. Line 262 "membrane-bound protein pump" should be "membrane-bound pump"
  8. Line Line 356-358. Tacrolimus and FK506 directly affect the ATPase activity of drug efflux pumps like Cdr1 in addition to acting on calcineurin and calmodulin so they act both directly and indirectly on drug efflux pump activity. Given the broad spectrum activity of fungal drug efflux pumps, many of the compounds discussed in this review may also be substrates of the drug efflux pumps and therefore compete with azole drugs for efflux. This concept should be noted.
  9. Bisphosphonates can be problematic due to dentally related problems and probably should be avoided.
  10. Line 362 The single sentence sentence description of the impact of proton pump inhibitors as synergiistic with azole drugs seems a little terse. It would be useful to include more of the back story to that observation i.e the plasma membrane proton pump in fungi is an essential enzyme that generates the membrane potential required for MFS transporter activity and the uptake of nutrients required for the energy production (ATP) that powers the ABC transporters that efflux azole drugs.
  11. Some of the panels in Figure 3 contains are difficult to interpret without referring to the explanation in the legend.  Can they be simplified using more appropriate arrows? Figure 3 g is misleading. In the figure the azole drugs appear to enter the cell via efflux pump rather than being effluxed. This is not correct.There are two different classes of azole transporting efflux pump in the plasma membrane - both should be drawn in the figure, and in particular the plasma membrane-bound ABC efflux pump should have a pair of intracellular ATP binding domains.
  12. Figure 1 CDC21 should be Cdc21

Reviewer 2 Report

Dear authors,

Thank you very much for this excellent article. If have some minor remarks below:

Table 1.

There seems to be problem with the layout of the table. The references should be in the first row of every antifungal drug.

In the row of isavuconazonium is a space between “mucormycosis” and “and” missing.

Figure 1 and 2.

These figures are beautiful! Unfortunately, the quality of figure 2 is not so good. Already in figure 1 the explanation in the grey box on the left bottom is difficult to read (mainly because of the size of the letters). In figure 2 the letters are almost unreadable, because the solution is too bad.

Table 2

Shouldn’t the category of f cyclosporin and tacrolimus be named calcineurin inhibitors?

Line 192: Fusarium spp.?

Line 219: There have been in vitro studies evaluating the interaction of colistin with isavuconazole against Aspergillius spp and against Candida auris. Recently colistin combined with fluconazole has shown to be synergistic against Candida albicans in vitro and increases survival in a Galleria mellonella infection model.

Line 346-358: Synergy of combinations of azoles with calcineurin inhibitors have also been demonstrated against filamentous fungi (Aspergillus and Mucorales spp.).

Table 3: The layout of table 3 got a bit messed up (see squared brackets). This should be corrected.

Reviewer 3 Report

This is a well-written and comprehensive review addressing an important issue.

Undoubtedly worth publishing in Pharmaceuticals.

No major comments

Minor points:

  1. Although this review is very extensive and covers almost all cases of reported synergistic effects between azole antifungals and other chemicals, the authors have missed synergism between azole antifungals and the chitin synthase inhibitor of natural origin Nikkomycin X/Z, reported in two independent papers in the years 1991/1992. The mechanism of this synergism seems quite interesting, so that it is worth mentioning.
  2. I would suggest changing the subtitle "2.2 Azole-Antibiotic Synergy" to "2.2 Azole-Antibacterial Synergy".  The term "antibiotic" usually refers to antimicrobial substances of natural origin, produced as secondary metabolites by microorganisms or their semi-synthetic derivatives. Some of the antibacterials mentioned in this chapter (sulfonamides, linezoid) are fully synthetic compounds. On the other hand, natamycin mentioned in the previous chapter is produced by Streptomyces, so that is an antifungal antibiotic.